Reliability and validity of My Jump 2® app to measure the vertical jump in visually impaired five-a-side soccer athletes

Silva Julio Cesar 1 juliociesar123@gmail.com
Silva Kalinne Fernandes 2
http://orcid.org/0000-0002-7068-276X Torres Vitor Bruno 2
Cirilo-Sousa Maria Socorro 3
Medeiros Alexandre Igor Araripe 4
Esmeraldo Melo Jacques Eanes 5
http://orcid.org/0000-0002-3294-8803 Batista Gilmário Ricarte 6
1 Department of Physical Education, State University of Ceará-UECE , Fortaleza, Ceará , Brazil
2 Federal University of Paraiba , João Pessoa, Paraíba , Brazil
3 Department of Physical Education, Regional University of Cariri , Crato, Ceará , Brazil
4 Institute of Physical Education and Sports, Federal University of Ceará , Fortaleza, Ceará , Brazil
5 State University of Ceará , Fortaleza, Ceará , Brazil
6 Department of Physical Education, Federal University of Paraíba , João Pessoa, Paraíba , Brazil
Jimenez Manuel
Electronic publication date: 2024 Oct 29
Publication date: 2024
Volume: 12
Electronic Location ID: e18170
Received 2024 May 1; Accepted 2024 Sep 3
Copyright: © 2024 Silva et al.
Copyright year: 2024
Copyright holder: Silva et al.
License: This is an open access article distributed under the terms of the Creative Commons Attribution License, which permits unrestricted use, distribution, reproduction and adaptation in any medium and for any purpose provided that it is properly attributed. For attribution, the original author(s), title, publication source (PeerJ) and either DOI or URL of the article must be cited.
License URL: https://creativecommons.org/licenses/by/4.0/

Keywords: Motion analysis, People with visual impairment, Physical performance, Testing, Athletes

Funding: The authors received no funding for this work.

==============================
Background

Although My Jump 2® consistently presented excellent values of reliability and validity when compared to force platforms (FPs) and contact mats, to date no scientific investigation assessed the validity and reliability of My Jump 2® to measure jump height in visually impaired five-a-side soccer athletes. Thus, the study aimed at analyzing the validity and reliability of the My Jump 2® to measure the vertical jump of five-a-side soccer athletes.

Methods

Twelve visually impaired five-a-side soccer athletes, volunteered for this study. Each player performed five countermovement jumps (CMJs) and squat jumps (SJs) on a contact platform (CP) while they were simultaneously recorded using My Jump 2®.

Results

There was almost perfect agreement between the My Jump 2® and the contact platform measurements of CMJ (intraclass correlation coefficient = 0.99; p < 0.001) and SJ (intraclass correlation coefficient = 0.99; p < 0.001), heights for athletes during the first and second measurement days. Bland-Altman analysis showed a bias of 0.25 ± 0.5 cm; maximum SD = 1.3; minimum SD = −0.88 for CMJ, while that Bland-Altman analysis showed bias 0.18 ± 0.5 cm; maximum SD = 1.3; minimum SD = −0.96, for SJ.

Conclusion

We can conclude that the My Jump 2® is a valid and reliable method to measure CMJ and SJ in visually impaired five-a-side soccer athletes.

Introduction

Force platforms (FPs) are considered the golden standard instruments to measure vertical jump height in athletes (Glatthorn et al., 2011; Requena et al., 2012). These platforms are able to measure the vertical jump height by using flight time (FT) methods (highly valid and reliable), and takeoff velocity (Moir, 2008). Nowadays, most instruments calculate jump height by measuring FT (Glatthorn et al., 2011; Moir, 2008; Requena et al., 2012).

Although FP, accelerometers, contact mats, infrared platforms and high-speed cameras (Casartelli, Müller & Maffiuletti, 2010; Glatthorn et al., 2011; Requena et al., 2012) have been validated to estimate vertical jump height through FT, these instruments have some disadvantages. The majority has shown to be not cost-effective for coaches and personal trainers and therefore they are used almost exclusively in academic laboratories and/or elite sport clubs. Additionally, these instruments are generally quite large and often need specific software for data analysis.

Over the last few years, portable instruments have been improving, but they still present some limitations that might interfere with their performance, namely in the field work. Consequently, a portable and low-cost approach to measure vertical jump performance was recently validated, My Jump 2® (Balsalobre-Fernández, Glaister & Lockey, 2015). This app combines the use of a high-speed and low-cost camera, and computer software to estimate vertical jump height. The My Jump 2® provides a cost-effective and portable alternative to traditional laboratory equipment, making it accessible and convenient for teams or organizations with budget constraints. The applications mobile-based platform allows for real-time monitoring and immediate feedback during training sessions, facilitating data-driven adjustments to training programs (Gençoğlu et al., 2023). In addition, countermovement jump (CMJ) and squat jump (SJ) height has been used to directly and indirectly assess muscle power (Rosas et al., 2016; Branquinho et al., 2020), bilateral asymmetry (Impellizzeri et al., 2007), the elastic properties of skeletal muscle and also as a tool to monitor neuromuscular readiness in training (Claudino et al., 2017). However, this approach was originally validated to assess recreationally active and healthy men (Balsalobre-Fernández, Glaister & Lockey, 2015; Driller et al., 2017), junior athletes (Rogers et al., 2019), sport sciences students (Carlos-Vivas et al., 2018; Haynes et al., 2019) and older people (Cruvinel-Cabral et al., 2018).

Hence, little attention has been given to paralympic sports, such as five-a-side soccer, which is played by visually impaired athletes (Leal et al., 2021). This sport is well-developed in Brazil and worldwide, and is one of the sports included in the Paralympic Games. Nevertheless, the scientific production with technology is scarce and the funding is low to support the sport for disabled people (DP) (Leal et al., 2021; Reis & Mezzadri, 2017). Therefore, examining the reliability and validity of My Jump 2® for the visually impaired athletes is important because visually impaired athletes have spatial-temporal, perceptual, and body control limitations (Santos et al., 2021; Finocchietti, Gori & Souza Oliveira, 2019).

Therefore, based on the validation the My Jump 2®, this technology can help several five-a-side soccer practitioners, including coaches and performance analysts, by enabling them to predict injury risks, facilitate talent identification, and seamlessly assess, monitor, and adapt their practices. Moreover, if My Jump 2® is revealed to be a reliable tool, it will enable the inclusion the CMJ and SJ assessments in five-a-side soccer throughout the sports season. Hence, this study aimed at analyzing the validity and reliability of the My Jump 2® to measure CM and SJ performance of five-a-side soccer athletes. The hypothesis of the study was that the My Jump 2® was valid and reliable for measuring the height of the CMJ and SJ of five-a-side soccer athletes.

Methods

Participants

Twelve elite male five-a-side soccer athletes (aged 28.5 ± 4.9 years; weight 72.0 ± 10.1 kg; height 172.2 ± 1.7 cm; with 7.0 ± 1.8 years of professional experience) participated in this study. Athletes were affiliated with the Brazilian Confederation of Sports for Visually Impaired Athletes (BCSV). The study comprised six paralympic champion athletes from the national five-a-side soccer team. The inclusion criteria were: (a) training frequency ≥ than three times a week; (b) at least 4 years of experience in five-a-side soccer, and (c) no recent muscle injuries that may interfere with the vertical jumps.

The study followed the guidelines stated in the Declaration of Helsinki and was approved by the Institutional Research Ethics Committee of Center University Unifacisa with the n. 5.882.073. Players were informed about the research scope, as well as the possibility to withdraw from the investigation at any time. Guarantees of confidentiality and anonymity were also explained. Afterwards, consent forms were signed by the participants.

Data collection instruments

To measure CMJ and SJ heights, we used a CP (600 mm × 900 mm) that recorded data at a 1,000 Hz frequency. The CP was connected to a computer with a software to analyze force data (Chronojump, version. 1.6.2; Boscosystem, Barcelona, Spain). This equipment has shown to be valid and reliable (De Blas et al., 2012; Pagaduan & Blas, 2013). An iPhone 11 pro max with an iPhone Operating System (iOS) (Apple, Cupertino, CA, USA) and the application called My Jump (version 2), which was designed and developed for iOS were also used.

Protocol of CMJ and SJ

CMJ and SJ height assessment (cm) was performed with a CP and an App compatible with iOS which was able to estimate jump height and FT through high–speed video recording that displays a high connection with CP (ICC = 0.997; p < 0.001) (Balsalobre-Fernández, Glaister & Lockey, 2015). Before CMJ and SJ measurements, athletes received instructions about movement execution and performed a familiarization test. CMJ: In a bipedal balance, both hands were positioned on the waist, and the athletes were instructed to squat and jump as high and fast as possible (Cruz et al., 2018). Five attempts were performed with 60-second rest. SJ: while wearing socks and sports shoes, the athlete adopted a 90° knee flexion position on the CP while uniformly distributing their body weight across both legs. When jumping, the athlete started from the squat position, with upright trunk, facing forward. The athlete performed a strong and fast extension of lower limbs without countermovement and while keeping hands on their hips. Athletes performed five jumps with 60-second rest.

Reliability Test to measure vertical jumps

The test-retest reliability of jump height using My Jump 2® was assessed in the afternoon with a seven-day interval between sessions. To obtain a reliable measure on test-retest of CMJ and SJ, the athletes were positioned upright at a 3-m distance from the evaluator. In each session, the athlete received verbal stimulus to perform the jump. Athletes performed five CMJs with 60-second rest intervals between attempts; and after a 10-minutes interval, performed five SJs with 60-second rest intervals. The same procedures were repeated during the retest.

Validation of performance

Validation was performed between My Jump 2® vs. CP to measure CMJ and SJ height (Hernández-Belmonte & Sánchez-Pay, 2021). The athlete was positioned on the CP and executed the jump while the evaluator was at a distance of three meters recording the jump with My Jump 2®. To validate the CP, the jump height was calculated from the impulse-moment theorem which allows the jumper’s takeoff speed to be calculated by double numerical integration of force. From the takeoff speed it is possible to determine jump height through the equation: Jump height = Voff2/2·g, where Voff is take-off speed and g is gravity acceleration (9.81 m/s2).

Statistical analysis

The distribuition of the normality was assessed using Shapiro-Wilk test. To analyze the reliability of the My Jump 2® for measuring jump height in comparison with the CP, the intraclass correlation coefficient (ICC) 2-way random single measures (consistency/absolute agreement) (2, 1) was used (Gallardo-Fuentes et al., 2016). Additionally, to analyze the stability of the app when measuring the five jumps executed for each exercise by each participant, Cronbach’s a and the coefficient of variation (CV) was used (Balsalobre-Fernández, Glaister & Lockey, 2015; Romero-Franco et al., 2017). The CV represented the typical error of measurements expressed as a percentage of the mean (Gallardo-Fuentes et al., 2016). To calculate the concurrent validity and interday reliability, the Pearson’s product moment correlation coefficient (r) was used. Finally, to complement the ICC analyses, Bland-Altman plot and systematic bias estimation and the 95% limits of agreement (±1.96 DP) were also used to analyze the agreement between My Jump 2® and CP (Bland & Altman, 1999; Hirakata & Camey, 2009). To analyze the proportion bias error was used linear regression. ICCs were reported with the flowing thresholds: >0.99, extremely high; 0.99–0.90, very high; 0.90–0.75, high; 0.75–0.50, moderate; 0.50–0.20, low; <0.20, very low (Buchheit & Mendez-Villanueva, 2013; Rogers et al., 2019). Data were presented as mean and standard deviation. Significance level was p ≤ 0,05. Data was analyzed through the Statistical Package for the Social Science (SPSS), version 25.0.

Results

There was almost perfect agreement between the My Jump 2® and the contact platform CMJ (ICC = 0.99; p < 0.001) and SJ (ICC = 0.99; p < 0.001), jump heights for male visually impaired athletes during the first and second measurement days (Table 1). Additionally, the Pearson’s product moment correlation coefficient showed almost perfect correlation between the My Jump 2® and the contact platform measurements for CMJ (r = 0.98, p < 0.001), SJ (r = 0.98–0.99, p < 0.001) jump heights for male visually impaired athletes during the first and second measurement days (Table 1).

Table 1 Jump performance measured with the My Jump 2® and the contact platform.

	First measurement day	Second measurement day	
	My Jump 2 ®	Contact platform	ICC	r	Mean difference (cm)	My Jump 2 ®	Contact platform	ICC	r	Mean difference (cm)	
CMJ (cm)	38.8 ± 3.5	38.5 ± 3.7	0.99	0.98	0.2 ± 0.5	38.5 ± 3.6	38.4 ± 3.5	0.99	0.98	−0.1 ± 0.5	
SJ (cm)	36.9 ± 3.2	36.7 ± 3.1	0.99	0.98	0.1 ± 0.5	37.1 ± 3.0	37.0 ± 2.9	0.99	0.98	−0.1 ± 0.5	
Note:

*ICC = intraclass correlation coefficient; r = Pearson’s product moment correlation coeficient.

The CMJ and SJ jump height values obtained from the My Jump 2® were not significantly different from the contact platform (p > 0.05). Bland-Altman analysis showed for CMJ the bias 0.25 ± 0.5 cm; maximum SD= 1.3; minimum SD= −0.88, while for SJ, Bland-Altman analysis showed bias 0.18 ± 0.5 cm; maximum SD= 1.3; minimum SD= −0.96 (Fig. 1).

Figure 1 Bland-Altman plots of CP and My Jump 2® (A) CMJ and (B) SJ height data.

The central line represents the absolute mean difference between the instruments, whereas the upper and the lower lines represent ±1.96 SD. R2 values shows that there was no proportion bias error.

In the analyses of test-retest reliability (i.e., intersession reproducibility) of the My Jump 2® and contact platform for all athletes, strong correlations were observed between the instruments for CMJ (contact platform: α = 0.91; CV= 9.1–9.6; r = 0.84; My Jump 2®: α = 0.91; CV= 9.0–9.3; r = 0.84), and SJs (contact platform: α = 0.90; CV= 7.8–8.4; r = 0.85; My Jump 2®: α = 0.90; CV= 8.0–8.4; r = 0.83) from the measurements obtained on day 1 and day 2 (Table 2).

Table 2 Intersession reliability of both the My Jump 2® and the contact platform.

	My Jump 2 ®	Contact platform	
	Intersession	Intersession	
	α	CV	r	α	CV	r	
CMJ (cm)	0.91	9.0–9.3	0.84	0.91	9.1–9.6	0.84	
SJ (cm)	0.91	8.0–8.6	0.84	0.90	7.8–8.4	0.85	
Note:

α, Cronbach’s alpha; CV, coeficient of variation; r, Pearson’s product moment correlation coeficient.

Discussion

This study investigated the reliability and validity of My Jump 2® to measure CMJ and SJ height of visually impaired five-a-side soccer athletes. The main findings were: (a) high reliability and agreement of CMJ and SJ height between CP and My Jump 2® and (b) no significant difference was found on CMJ and SJ height between FP and My Jump 2®.

The importance of this App to the sports practice is evident in literature, however, most of the studies that validated the instrument performed only CMJ (Balsalobre-Fernández, Glaister & Lockey, 2015; Carlos-Vivas et al., 2018; Cruvinel-Cabral et al., 2018). Therefore, the findings of the present study broaden the use of My Jump 2® to visually impaired five-a-side soccer athletes, although this study did not measure other sports performance, the results suggest that high reliability could also be achieved when measuring other sports performances.

Pearson’s Correlation Coefficient showed an almost perfect correlation between My Jump 2® and the measurement of CP in CMJ height of five-a-side soccer athletes (0.98; p < 0.001). Balsalobre-Fernández, Glaister & Lockey (2015) found the same in an investigation conducted with recreationally active and healthy male students. The authors verified the agreement of CMJ measure between CP and My Jump 2® App (0.995; p < 0.001) and also Cruvinel-Cabral et al. (2018) found the same in a research with older people (0.999; p < 0.001).

Despite the differences between CP sampling frequency (1,000 Hz) and My Jump 2® (240 Hz), CMJ and SJ height values were quite similar between both methods, as seen in ICC. The findings reflect an almost perfect comparison between My Jump 2® and FP. The higher value of this correlation compared to Balsalobre-Fernández, Glaister & Lockey (2015) and Cruvinel-Cabral et al. (2018) might be explained due to the iPhone updated version, which has a more efficient camera that allows a video with more squares enabling a higher sampling frequency. Thus, the selection of takeoff starts and landing phases is more accurate because estimated values from different devices are nearer in relation to the same jump (Balsalobre-Fernández, Glaister & Lockey, 2015).

Moreover, it is important to note that although the My Jump 2® requires the time of flight to calculate jump height (Balsalobre-Fernández, Glaister & Lockey, 2015). The force platforms are often considered the gold standard due to their ability to calculate jump height based on the impulse-momentum theorem, which takes into account the total force applied during the jump and the duration of this force (Gençoğlu et al., 2023). However, it is clear from the literature that time-of-flight calculations are also commonly used on force platforms. For this reason, the values of the CMJ and SJ performed by visually impaired athletes showed a good level of agreement between the devices. Therefore, in scenarios where the device, and not the method, serves as a reference for jumping performance, this study showed that My Jump 2® appears to be a viable alternative, because it provides a cost-effective and portable alternative to traditional laboratory equipment, making it accessible and convenient for teams or organizations with budget constraints.

Although the five-a-side soccer athletes are visually impaired have spatial-temporal, perceptual and body control limitations (Santos et al., 2021), the visually impaired five-a-side soccer players who took part in this study are considered to be high-performance athletes who constantly perform these types of muscle power assessments and were very familiar with the CMJ and SJ tests. Thus, this impairment did not interfere in the performance of CMJ and SJ in the study.

Although CMJ and SJ performance represents an “explosive” muscle action, some technical actions in five-a-side soccer, e.g., “ball possession” and the “defense action” require a sub-maximal high speed, which elicits physiological and biomechanical variables similar to those found in jumping, such as optimal neuromuscular characteristics related to reflex and voluntary neural activation, strength, muscle elasticity and anaerobic characteristics (Bulbulian, Wilcox & Darabos, 1986; Gallardo-Fuentes et al., 2016). However, when using My Jump 2®, it is important for coaches to carry out jump familiarization sessions with visually impaired athletes, so that they have reliable data when assessing CMJ and SJ height.

Five-a-side soccer is a Paralympic modality with a high visibility in sports for people with disability, but many clubs do not have financial resources to acquire golden standard equipment, e.g., FPs or contact mats. Therefore, My Jump 2® presents as an alternative capable of measuring with accuracy vertical jump height (does not demand previous experience of evaluators in video analysis) targeting the majority of the population, including trained athletes. This is important considering that, vertical jump performance is a fundamental characteristic among the different sports, either for adults (Andersson et al., 2010; Brumitt et al., 2014) and young people (Ramirez-Campillo et al., 2019). Considering that vertical jump tests might be a pattern to assess muscle power (Claudino et al., 2017), it would be important to conduct these tests throughout the training and competitive seasons of five-a-side soccer athletes with assessment tools that allow easy, accurate, valid, reliable and low-cost measurements.

My Jump 2® will enable physical conditioners to use this tool in physical evaluations to indirectly analyse muscle power and bilateral asymmetry and monitor athletes’ neuromuscular readiness before training sessions. Consequently, the training might facilitate injury prevention. This study has a few limitations that warrant further discussion. A methodological limitation of My Jump 2® to measure jump height, mentioned in previous studies by Cruvinel-Cabral et al. (2018), is the fact that the evaluator must manually select the photograms when the individual performs the take-off and landing moments, which makes the measurement process subjective and slow in some cases.

Conclusion

We concluded that My Jump 2® is a valid tool for measuring CMJ and SJ on visually impaired five-a-side soccer athletes. In addition, our findings could encourage trainers and coaches, to evaluate muscle power and monitor the elastic properties of skeletal muscle and neuromuscular readiness status, which is relevant information to improve decision making in training control and prescription of exercise. Further research should validate this innovative technological instrument for female athletes.

Supplemental Information

Supplemental Information 1 Data validation of my jump app.

The authors would like to thank the visually impaired five-a-soccer athletes who volunteered for the study.

Additional Information and Declarations

Competing Interests

Author Contributions

Human Ethics

Data Availability

Alexandre Igor Araripe Medeiros is an Academic Editor for Peerj.

Julio Cesar Silva conceived and designed the experiments, performed the experiments, analyzed the data, prepared figures and/or tables, authored or reviewed drafts of the article, and approved the final draft.

Kalinne Fernandes Silva conceived and designed the experiments, performed the experiments, authored or reviewed drafts of the article, and approved the final draft.

Vitor Bruno Torres conceived and designed the experiments, authored or reviewed drafts of the article, and approved the final draft.

Maria Socorro Cirilo-Sousa conceived and designed the experiments, authored or reviewed drafts of the article, and approved the final draft.

Alexandre Igor Araripe Medeiros conceived and designed the experiments, analyzed the data, prepared figures and/or tables, authored or reviewed drafts of the article, and approved the final draft.

Jacques Eanes Esmeraldo Melo conceived and designed the experiments, authored or reviewed drafts of the article, and approved the final draft.

Gilmário Ricarte Batista conceived and designed the experiments, authored or reviewed drafts of the article, and approved the final draft.

The following information was supplied relating to ethical approvals (i.e., approving body and any reference numbers):

The study followed the guidelines stated in the Declaration of Helsinki and was approved by the Institutional Research Ethics Committee of Center University Unifacisa with the n. 5.882.073.

The following information was supplied regarding data availability:

The data are available in the Supplemental File.

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
