# Peer review of "Reliability and validity of My Jump 2® app to measure the vertical jump in visually impaired five-a-side soccer athletes"

_PeerJ, doi:10.7717/peerj.18170_

## Round 0.1 · original submission · Major Revisions

Please, respond to every reviewer comment

Dr. Manuel Jiménez López.

Reviewer 1 ·

Basic reporting

The structure, writing, and presentation of the article are adequate, as well as the bibliographic references used based on the study's theme.

Experimental design

The study is innovative, the research has been thoroughly described and analyzed, and ethical aspects have been considered. The methods and procedures are clear.

Validity of the findings

No comment

Reviewer 2 ·

Basic reporting

Whole paper: Please add “the” before “My Jump 2.”

Line 18: Please modify to “vertical jump of five-a-side.”

Line 23: Please delete “jumps” following heights.

Line 23, 26: Please modify to “showed a bias of XXX for CMJ or SJ.

Line 46, 59, 78, and 79: Please add a comma before “and.”

Line 48: Please revise “The majority have.”

Lines 51, 56: Please delete “a” before specific software.

Line 52: Please correct to “Over the last few years.”

Line 55: Please correct to “My Jump 2.”

Line 58: Please modify to “sport science.”

Line 60: Please delete “to” five-a-side...

Line 61: Please modify to “visually.”

Line 64: Please add “the” before literature.

Lines 79-80: Please change the form of the verb: My jump 2 is revealed to be...

Line 82: Please modify to “the CMJ and SJ.”

There are many minor errors in other sections as well. Please review them again.

I did not understand whether by targeting visually impaired individuals, the reliability and validity change or not compared to the other previous studies. Cruvinel-Cabral et al. (2018) hypothesized that the reliability and validity might decrease due to the lower jumping height of older adults. Thus, if evidence does not show that jump height or characteristics affecting jumping height, such as movement in visually impaired individuals, differ from those in other populations, the significance of this study will not be understood.

Lines 183-187: This passage seems to be redundant as it has already been mentioned in the Introduction section. It can be omitted.

Lines 192 and 234: Why can bilateral asymmetry be detected through SJ? Since it involves jumping with both legs, it may seem like it wouldn't detect any asymmetry. Please explain logically how it can be detected. In the study by Claudio et al. (2016), there is no mention of asymmetry between the left and right sides.

Experimental design

No comment.

Validity of the findings

No comment.

Additional comments

I could not understand the significance of verifying reliability and validity again in this study using equipment that has already been adequately investigated. Without mentioning the potential errors specific to visually impaired individuals when performing jumps, it may be difficult to understand the novelty.

·

Basic reporting

The study aimed to validate the My Jump2 app as a reliable instrument for evaluating football athletes, the paper approaches an interesting theme and has been well-written and well-organized. The introduction brings an explanation of basic concepts and the literature concerning the study question.

Experimental design

no Comment

Validity of the findings

No comment

Additional comments

I suggest exploring more in introducing practicality in the use of mobile applications in relation to other equipment from both a financial point of view and ease of field use.

·

Basic reporting

Overall a good research study, with appropriate methods and analyses. Below there are some detailed points.

Language – Language seems fine and to the point.

References are appropriated, article structure and data are clear through the article.

Experimental design

Introduction

The introduction is well written leading to problem, however I believe it still need to highlight why this research is necessary. Although the first paragraph of the second page shows the importance of the population and scarce information regarding them, I suggest to insert some points of why the jumps could be different for the visual impaired athletes, thus explaining why this research is needed.

Experimental Design

The experimental design is simple and well stated

Methods

I suggest at line 123 to standardize the number (five CMJ and 5 SJ to 5 and 5 or five and five), also to avoid confusion change 10m to 10 minutes.

Validity of the findings

Statistics seem appropriated and broad enough to get valid findings. However, I could not find the information regarding which data was used: the best performed jump or the average of the five jumps? This should be clarified at statistics or earlier in the methods section.

Results are clear, straight to the point and seem valid

Discussion

Lines 182-183, I suggest a change to the my jump2 app has never been used so far in published research article to assess... - although it seems to much detail, I believe avoids confusion and reinforce that the findings may be important to the performance field.

I also suggest to make the discussion more robust to have discussion to why visually impaired athletes could have any differences ate their jumps. And another suggestion is to discuss the data coming from different analyses – flight time (My Jump 2 App) and Takeoff Velocity (CP).

Conclusion

Conclusion is to the point, how it should be. But I still make a suggestion to explain in what extend this research is valid ( for example muscle power, strength and elastic components, monitoring) and important to the population – nothing different to what was stated at the discussion.

---

## Round 0.2 · accepted · Accept

Dear Authors:

Sorry for the delay. I needed to wait for a final review, I apologize. But, your manuscript "Reliability and validity of my jump 2® app to measure the vertical jump in visually impaired five-a-side soccer athletes" has been already accepted for publication.

Thank you for your patience

Dr. Manuel Jiménez

Reviewer 1 ·

Basic reporting

The authors have addressed all the suggested revisions, and there are no further comments or recommendations at this time.

Experimental design

The authors have addressed all the suggested revisions, and there are no further comments or recommendations at this time.

Validity of the findings

The authors have addressed all the suggested revisions, and there are no further comments or recommendations at this time.

Additional comments

The authors have addressed all the suggested revisions, and there are no further comments or recommendations at this time.

Reviewer 2 ·

Basic reporting

Unfortunately, there are numerous errors throughout the manuscript, as mentioned in my first review, making it difficult to focus on the content. I strongly recommend a thorough revision.1) Please unify the terms used throughout the manuscript.

1. Consistency of Terminology: Please standardize the terms used throughout the manuscript. Examples include:
• Variations such as "my jump 2® app," "My Jump2®," "My (not italic) Jump 2®," "My Jump2® App," and " My Jump 2® app," and “my jump app” should be unified.
• The terms “CMJ and SJ” or “CM and SJ” should be consistently used.
• Use either “60s” or “60-second” uniformly.
• Standardize capitalization, such as "Correlation Coefficient" or "correlation coefficient."
2. Spacing Issues: Double spaces appear in several locations (e.g., lines 63, 65, 89, 91, 96, 250, 272, 273, and 274). Please correct these.
3. Formatting: Ensure consistently use symbols like “=” and “±”. Examples include:
"ICC = 0.99"
"SD= 1.3" (no space before “=”)
"p≤0,05" (no space and using a comma instead of a period)
"28.5 ± 4.9" and "±1,96" (no space and using a comma)
4. Decimal Points: Use periods (.) for decimal points instead of commas.
5. Abbreviation: Do not abbreviate terms at their first appearance. Please define "CMJ" and "SJ," and confirm where “ICC” first appears in the manuscript.
6. Significant Digits: Ensure uniformity in the significant digits (e.g., consistently use "r = 0.999").
7. Punctuation: Add a period at the end of the sentence on line 223.
8. Readability: Please refer to the following minor comments.
9. There are many errors in the use of articles.

I also noticed discrepancies between the PDF and Word files of the manuscript. My review is based on the PDF file.

・Minor comments
Line 50: I suggest revising the phrase to a valid and reliable method to measure CMJ and SJ “height” for accuracy.
Line 78: Remove “a” before “computer software.”
Line 129: The word “than” is unnecessary.
Line 134: Replace “players” with "athletes" or "participants," as "players" has not been used elsewhere in the manuscript.
Line 139: The word “height” is repeated in one sentence; please remove the first occurrence.
Line 140: Please modify “strength data” to “force data”
Line 143: Please describe the version of the application.
Line 146: Please revise “CMJ and SJ height assessment (cm) was performed” to “CMJ and SJ height was assessed.”
Lines 150 & 152: Why are CMJ and SJ written in their full names (also in Line 196 contact platform)?
Line 151: Please revise to “and the athletes were instructed...”
Lines 154 & 156-157: The explanation about placing hands on the hips is mentioned twice, so please remove one of them.
Line 159-160: The sentences were not clear, and thus please improve the readability. For example, “The test-retest reliability of jump height using My Jump 2® was assessed (measured) in the afternoon with a seven-day interval between sessions.”
Line 174: Please modify the description of the equation to improve the readability.
Jump height = Voff2/2・g, where Voff is take-off speed and g is gravity acceleration (9.807 m/s2).
Line 177: Please revise to “The distribution of the normality was assessed using the Shapiro-Wilk test.”
Lines 197 & 201: Please modify “heights jumps” to “jump heights.”
Line 205: Please delete CMJ and SJ “jump” height.
Line 221: Please revise CMJ and SJ performance to CMJ and SJ height.
Lines 240–241: This sentence might be difficult to read. I suggested revising the sentence to “The present study obtained the higher correlation coefficients as compared to previous studies (Balsalobre-Fern·ndez et al. and Cruvinel-Cabral et al.). This would be (is likely) because...”
Line 250: Which literature is being referred to? Is it the current study or the one by Gencoglu et al.?

・Major comments
The introduction section appears to be somewhat redundant. As a suggestion, paragraphs 5 ("This research is ...") through 7 ("Therefore, during sporting activity...") could be omitted from the introduction to enhance conciseness. Here is a suggestion in the attached file for your reference. This is just my opinion, so you don't need to make changes exactly as suggested. Also, please describe your hypothesis.

This study did not assess the reliability and validity of muscle power, bilateral asymmetry, the elastic properties of skeletal muscle, and neuromuscular readiness status. Thus, you cannot lead to the suggestion (Lines 227–231). You should soften the expression. For example, “although this study did not measure other sports performances, the results suggest that high reliability could also be achieved when measuring other sports performances. Future studies...”

Line 232: I could not catch what you want to discuss in the 3rd paragraph (Pearsonís Correlation Coefficient showed...) of the discussion section. What is your message in this paragraph?

Line 246:I found the main point of the 5th paragraph difficult to understand. It was unclear to me how it logically leads to the sentence, "For this reason, the values of the CMJ and SJ jumps performed by visually impaired athletes showed a good level of agreement between the devices." I believe a concise and clear explanation is needed to make this connection.

Line 257: Why is it necessary to compare the jump heights of male visually impaired five-a-side soccer athletes with those of male and female track and field athletes? Given the differences in the populations, it is not possible to accurately discuss whether the values of the jump heights are comparable. The discussion on whether the movements are performed correctly or whether the presence of a disability affects the jumping motion is not possible without comparing the jumping motions of visually impaired and non-impaired individuals participating in the same sport. Additionally, this discussion does not seem relevant to the main text, so I would recommend removing it.

Line 266: This paragraph makes no sense at all. I don't understand why the Gallardo-Fuentes et al. paper was referenced. Also, why is the need for explosive movements in five-a-side soccer mentioned? What is the main point of this paragraph? The sentences are too long, which further obscures the message.

Line 277: What results is this discussion based on? Discussions should be conducted in response to the results. A similar message is found in the introduction, so shouldn't it be mentioned there instead?

Experimental design

You described “wearing socks or barefoot (Line 152)” when measuring the SJ height. Could you clarify whether the socks were worn (or barefoot) throughout the measurement, or were the shoes only removed for the SJ?

Validity of the findings

Line 291: Wouldn't this limitation be more accurately described as a flaw in the experiment rather than just a limitation? What was the reason that this measurement could not be performed? Is it impossible to measure it from now on?

Additional comments

Again, there are numerous errors throughout the manuscript. I strongly recommend a thorough revision.

Annotated reviews are not available for download in order to protect the identity of reviewers who chose to remain anonymous.

·

Basic reporting

no comment

Experimental design

no comment

Validity of the findings

no comment

Additional comments

The authors answered all of my comments, and they made appropriate changes in the manuscript. Therefore I recommend this manuscript be published,

·

Basic reporting

The authors aimed to answer all questions and to improve the article. It is clear the changes in the text.

Experimental design

The experimental design is to the point, described appropriately. The authors aimed to improve why this research is relevant.

Validity of the findings

The findings are validity and meaningful to the population investigated, which was improved by the authors in this version.

Additional comments

The authors made all the changes regarding to answer all the reviewers doubts. The methods and conclusion are in place to the findings.